

# Using the FRAIL scale to compare pre-existing demographic lifestyle and medical risk factors between non-frail, pre-frail and frail older adults accessing primary health care: a cross-sectional study

Vanessa Aznar-Tortonda, Antonio Palazón-Bru and
Vicente Francisco Gil-Guillén

Department of Clinical Medicine, Miguel Hernández University, San Juan de Alicante, Spain

## ABSTRACT

**Background:** Few studies in the scientific literature have analyzed frailty status as an ordinal variable (non-frail, pre-frail and frail) rather than as a binary variable (frail vs non-frail). These studies have found that pre-frailty behaves differently from frailty (no ordinality in the variable). However, although the comparison between pre-frail and frail individuals is clinically relevant to understanding how to treat pre-frailty, this comparison was not performed in previous studies.

**Materials and Methods:** A cross-sectional observational study was designed with 621 older individuals aged ≥60 years in Spain in 2017–2018, determining factors associated with a higher frailty stage (non-frail, pre-frail and frail) and undertaking this comparison, in addition to measuring non-frailty. The factors assessed through a multinominal regression model were: age, sex, living alone, recent loss of the partner, income and total comorbidities.

**Results:** Of the total participants, 285 were non-frail (45.9%), 210 were pre-frail (33.8%) and 126 were frail (20.3%). Compared to non-frail individuals, pre-frail individuals were older, with more comorbidities and a lower income. Compared to non-frail individuals, frail individuals were more likely to be female, older, with more comorbidities and a lower income. Compared to pre-frail individuals, frail individuals were more likely to be female, older and with more comorbidities.

**Conclusion:** Comparison between the pre-frail and frail groups showed that frail persons were more likely to have a lower income, be female, older and have a higher number of comorbidities.

Corresponding author
Antonio Palazón-Bru,
antonio.pb23@gmail.com

## INTRODUCTION

Frailty is a biological syndrome (*Dent et al., 2019*), characterized by a state of vulnerability and increased susceptibility to stressful factors due to loss of homeostasis after a stressor event, resulting in the cumulative deterioration of multiple physiological systems (*Clegg et al., 2013*; *Hoogendijk et al., 2019*). Frailty is a dynamic state that is not

unidirectional, with persons shifting in a continuum, non-frail–pre-frail–frail (*Abizanda Soler & Rodríguez Mañas, 2014*). Frail older people have decreased physiological reserves, which increases their vulnerability and likelihood of adverse health episodes. This is a transition from non-frailty to frailty and then to dependance (*Rockwood et al., 1994*; *Whitson, Purser & Cohen, 2007*; *Dent, Kowal & Hoogendijk, 2016*). The last 30 years have seen great but unsuccessful efforts to find a standard instrument to identify frailty (*Hoogendijk et al., 2019*). The most used of the many tools to measure frailty is the Phenotype model, described by Fried in 2001. This contemplates five variables: unintentional weight loss, self-reported exhaustion, low energy expenditure, slow gait speed, and weak grip strength. Older adults with three or more factors are considered frail, with one or two factors pre-frail, and with no factors non-frail (*Fried et al., 2001*; *Clegg et al., 2013*). Another tool is the FRAIL scale (Fatigue, Resistance, Ambulation, Illnesses, and Loss of Weight), which considers individuals to be frail if three or more criteria on the scale are met, pre-frail if one or two criteria are met, and non-frail if none are met (*Van Kan et al., 2008*). The FRAIL scale is a brief simple questionnaire comprising five items. It requires no special material or measurements and can therefore be done quickly to identify the state of frailty (*Kojima, 2018*). It has been validated in several populations and is increasingly used in clinical and research settings (*Hoogendijk et al., 2019*).

The Phenotype model of Fried, which has been independently validated, can form the basis for the detection of frailty in daily clinical practice, though it is still not clear how the variables of the definition can be reliably translated (*Clegg et al., 2013*). Many other models to measure frailty are also available, such as: Cumulative deficit model, Clinical Frailty Scale, gait speed measurement, the Groningen Frailty Indicator and the Edmonton Frail Scale (*Clegg et al., 2013*; *Hoogendijk et al., 2019*).

Most medical studies have assessed the presence or absence of frailty, without evaluating pre-frailty (*Carneiro et al., 2017*). However, recent longitudinal studies have determined that 8.2–18.2% of older adults in pre-frailty progress to frailty over a mean period of approximately 5 years (*Doi et al., 2018*; *Herr et al., 2019*; *Kojima et al., 2019*), with the risk of this transition increasing with exhaustion, physical inactivity, decreased muscle strength and decreased mobility (*Doi et al., 2018*). Many studies have assessed the factors associated with both frailty and pre-frailty (*Abizanda et al., 2011*; *Hoogendijk et al., 2016*; *Langholz et al., 2018*; *Jacobsen et al., 2019*), determining whether there is a trend between a greater degree of frailty and each individual factor studied (trend test), that is, considering the stage of frailty to be an ordinal variable. This, however, does not enable us to draw conclusions about possible differences between the stages of frailty. This can be done by comparing the groups independently (e.g., pre-frail vs non-frail) using independent statistical models (binary logistic regression models) or a single model assessing the three possible stages jointly (multinomial or ordinal logistic regression models), that is, without leaving out information. Four cross-sectional studies have analyzed frailty status as an ordinal variable (non-frail, pre-frail and frail) rather than as a binary variable (frail vs non-frail) and have studied differences between the groups; one from the United Kingdom (UK), another from Sri Lanka, another from China and the final one from Colombia (*Curcio, Henao & Gomez, 2014*; *Hanlon et al., 2018*; *Ye, Gao & Fu, 2018*;

*Siriwardhana et al., 2019*). The first two studied whether this ordinality made sense through ordinal and multinomial regression models and saw that the latter have a better fit, that is, pre-frailty behaves differently than frailty, so it should be considered independently. However, neither study compared pre-frail vs frail individuals, and it is important to ascertain whether different patterns exist between the two stages to consider how to treat a pre-frail person. For this reason, we conducted a study with older people aged 60 or over using a multinomial logistic regression model to determine whether different pre-existing demographic, lifestyle and medical risk factors exist between non-frailty, pre-frailty and frailty (*Curcio, Henao & Gomez, 2014*; *Hanlon et al., 2018*; *Siriwardhana et al., 2019*), focusing especially on the comparison between frail and pre-frail individuals. Although the group of pre-frail persons could be considered to have less importance as they have received less attention in previous studies, their prevalence is not small (*Curcio, Henao & Gomez, 2014*; *Hanlon et al., 2018*; *Ye, Gao & Fu, 2018*; *Siriwardhana et al., 2019*). Bearing in mind that over a period of 5 years between one in five and one in eight persons pass from pre-frail to frail (*Doi et al., 2018*; *Herr et al., 2019*; *Kojima et al., 2019*), knowing the differences between these groups has great clinical relevance, as it would provide information that can then be used to prevent or delay this situation.

## MATERIALS AND METHODS

### Study population, design, participants and ethical considerations

The study population included individuals aged 60 or over who visited primary care centers in the Valencian Community (Spain). Health care in Spain is universal and free of charge for all residents.

This cross-sectional observational study analyzed a sample of patients who sought care between 2017 and 2018 without an appointment at several primary healthcare centers in the Valencian Community: Sagunto and Puerto de Sagunto (Valencia), Monóvar, Las Acacias and Marina Española (Alicante). All patients aged 60 or over visiting these healthcare centers were invited to participate in the study, excluding those who did not expressly wish to participate or who had any physical, psychological or social problem that prevented them from completing the questionnaires that would later be used (see "Variables and Measurements").

The Clinical Research Ethics Committee of Sagunto Hospital and Elda University General Hospital approved the study on March 6, 2017. All participants in the study gave their informed consent in writing.

### Variables and measurements

The main study variable was frailty status, which can have three stages in older people: non-frail, pre-frail and frail. This is evaluated with the FRAIL frailty scale, assessing the number of criteria presented by each patient. Each of the criteria (fatigue, resistance, ambulation, illness, and loss of weight) was obtained through a personal interview (*Fried et al., 2001*; *Ferrucci et al., 2004*; *Rolfson et al., 2006*).

Secondary variables assessed to determine whether they were risk factors for a higher frailty stage were: age, sex, living alone, recent loss of the partner (less than 1 year), income

level (< minimum wage (MW), MW-1.5 × MW and >1.5 × MW; the minimum wage was approximately 630 euros nationally at the time of data collection) and total number of comorbidities. The sum of the number of comorbidities included: stroke, coronary heart disease, arthrosis or advanced osteoarticular disease, depression, diabetes and chronic obstructive pulmonary disease because these have shown associations with a higher level of frailty (*Hanlon et al., 2018*). The justification for selecting these potential factors is that in the three studies assessing frailty as a continuum with three stages (*Hanlon et al., 2018*; *Ye, Gao & Fu, 2018*; *Siriwardhana et al., 2019*), associations were found with age, sex, marital status and cohabiting. Finally, age was evaluated in a binary manner, setting 80 years as the cut-off point, since it appears in the definition of frailty (*Rockwood et al., 1994*; *Whitson, Purser & Cohen, 2007*; *Dent, Kowal & Hoogendijk, 2016*), in addition to being used in previous studies (*Siriwardhana et al., 2019*), as this is the most frail age group.

### Sample size calculation and statistical analysis

The sample size was calculated to estimate the prevalence of pre-frailty and frailty in the older population aged 60 years and older. Expecting to find 39.5% and 16.9% of patients with these conditions (*Ye, Gao & Fu, 2018*), respectively, setting type I error at 5% and with an accuracy of 4%, at least 574 subjects in total would be needed for pre-frailty and 338 for frailty. Consequently, with 574 patients we could estimate the two proportions with the established error.

The qualitative variables were described using absolute and relative frequencies. Income was analyzed as a qualitative variable with the lowest income bracket being set as the reference point. For the single quantitative variable (number of comorbidities), means and standard deviations were calculated if normally distributed, otherwise we used median and interquartile range. The Jonckheere–Terpstra test was used to determine associations between our secondary variables and the main variable (frailty status), the latter being ordinal. We then performed a post-hoc analysis using the Bonferroni correction with the Pearson Chi squared and median tests. Using a multivariate approach, a multinomial rather than an ordinal logistic regression model was estimated as the literature has shown these to have a different character concerning the behavior of associated factors (*Hanlon et al., 2018*; *Siriwardhana et al., 2019*). With this model, the three groups were compared through analysis of the adjusted odds ratios for all the secondary variables. The type I error was 5% for all analyses, and for each relevant parameter its associated confidence interval (CI) was calculated. The IBM SPSS Statistics 25 statistical package was used for all the calculations.

## RESULTS

Of the 621 older patients included in the study (all those invited agreed to participate), 285 were non-frail (45.9%, 95% CI [42.0–49.8]), 210 were pre-frail (33.8%, 95% CI [30.1–37.5]) and 126 were frail (20.3%, 95% CI [17.1–23.5]). Regarding the secondary variables (Table 1), in terms of demographic characteristics, it should be noted that there was a greater proportion of women (58.6%) and that approximately one in five individuals was

**Table 1 Descriptive and bivariate analysis of the factors associated with frailty in the older individuals.**

| Variable | Total<br>n = 621<br>n (%)/median (IQR) | Non-frail<br>n = 285 (45.9%)<br>n (%)/median (IQR) | Pre-frail<br>n = 210 (33.8%)<br>n (%)/median (IQR) | Frail<br>n = 126 (20.3%)<br>n (%)/median (IQR) | p-value* |
|---|---|---|---|---|---|
| Female sex | 364 (58.6) | 148 (40.7) | 125 (34.3) | 91 (25.0) | <0.001 |
| Age >80 years | 133 (21.4) | 38 (28.6) | 48 (36.1) | 47 (35.3) | <0.001 |
| Living alone | 103 (16.6) | 34 (33.0) | 41 (39.8) | 28 (27.2) | 0.004 |
| Recent loss of the partner | 11 (1.8) | 3 (27.3) | 6 (54.5) | 2 (18.2) | 0.391 |
| Income:<br>Low (<MW)<br>Medium (MW-1.5 × MW)<br>High (>1.5 × MW) | 187 (30.1)<br>228 (36.7)<br>206 (33.2) | 71 (38.0)<br>97 (42.5)<br>117 (56.8) | 69 (36.9)<br>87 (38.2)<br>54 (26.2) | 47 (25.1)<br>44 (19.3)<br>35 (17.0) | <0.001 |
| Number of comorbidities | 1 (2) | 1 (1) | 1 (2) | 2 (1) | <0.001 |

Notes:
   IQR, interquartile range; MW, minimum wage; n(%), absolute frequency (relative frequency). Comorbidities included: stroke, coronary heart disease, arthrosis or advanced osteoarticular disease, depression, diabetes and chronic obstructive pulmonary disease.
   * Jonckheere–Terpstra test.

**Table 2 Post-hoc with the Bonferroni correction of the bivariate analysis of the significant (p < 0.05) factors associated with frailty in the older individuals (p-values).**

| Variable | Pre-frail vs non-frail | Frail vs non-frail | Frail vs pre-frail | Number of comparisons | Threshold p-value for significance |
|---|---|---|---|---|---|
| Female sex | 0.093 | <0.001 | 0.014 | 3 | 0.017 |
| Age >80 years | 0.006 | <0.001 | 0.004 | 3 | 0.017 |
| Living alone | 0.020 | 0.007 | 0.553 | 3 | 0.017 |
| Income:<br>Low (<MW)<br>Medium (MW-1.5 × MW)<br>High (>1.5 × MW) | 0.052<br>0.093<br><0.001 | 0.010<br>0.862<br>0.010 | 0.407<br>0.236<br>0.678 | 9 | 0.0056 |
| Number of comorbidities | <0.001 | <0.001 | <0.001 | 3 | 0.017 |

Notes:
   MW, minimum wage.
   Comorbidities included: stroke, coronary heart disease, arthrosis or advanced osteoarticular disease, depression, diabetes and chronic obstructive pulmonary disease.

over 80 years of age (21.4%). In addition, one in six individuals lived alone (16.6%), and 30.1% had an income below the minimum wage. Finally, the mean number of comorbidities was 1.1.

Concerning the factors associated with frailty status as a continuum (higher likelihood of being in a more advanced stage of frailty) (Table 1), significant differences ($p < 0.05$) were observed in all the variables, with the exception of recent loss of the partner ($p = 0.391$). These differences were due to the frailty stage being more advanced in people of older age, women, those living alone at home, and those with a lower level of income. Finally, the number of comorbidities increased as the frailty stage was higher. Analysis within each group of these differences (post-hoc analysis, Table 2) showed that female sex differed between the frail group and the other two groups, whereas living alone differed between the frail and the non-frail groups. Age and number of comorbidities differed between the three groups, whereas income just differed between the non-frail and the pre-frail groups in the category of higher amounts.

Table 3 Multinomial logistic regression of frailty status among the older individuals.

| Variable | Pre-frail vs non-frail Adj. OR (95% CI) | p-value | Frail vs non-frail | p-value | Frail vs pre-frail | p-value |
|---|---|---|---|---|---|---|
| Female sex | 1.05 [0.71–1.57] | 0.804 | 1.89 [1.11–3.20] | 0.019 | 1.79 [1.06–3.03] | 0.029 |
| Age >80 years | 1.70 [1.04–2.78] | 0.036 | 3.16 [1.79–5.56] | <0.001 | 1.86 [1.10–3.13] | 0.020 |
| Living alone | 1.65 [0.96–2.84] | 0.069 | 1.73 [0.90–3.31] | 0.099 | 1.05 [0.58–1.91] | 0.880 |
| Recent loss of the partner | 1.13 [0.26–4.92] | 0.870 | 0.44 [0.07–3.03] | 0.408 | 0.39 [0.07–2.11] | 0.276 |
| Income: | | | | | | |
| Low (<MW) | 1 | 0.524 | 1 | 0.042 | 1 | 0.109 |
| Medium (MW-1.5 × MW) | 0.86 [0.54–1.37] | 0.002 | 0.54 [0.30–0.98] | 0.005 | 0.63 [0.36–1.11] | 0.817 |
| High (>1.5 × MW) | 0.45 [0.27–0.74] | | 0.42 [0.22–0.77] | | 0.93 [0.50–1.72] | |
| Number of comorbidities | 1.79 [1.43–2.25] | <0.001 | 3.12 [2.39–4.08] | <0.001 | 1.74 [1.37–2.21] | <0.001 |

Notes:
Adj. OR, adjusted odds ratio; CI, confidence interval; MW, minimum wage (€630).
Comorbidities included: stroke, coronary heart disease, arthrosis or advanced osteoarticular disease, depression, diabetes and chronic obstructive pulmonary disease.

When analyzing the results obtained in the multinomial model (Table 3), it is important to note that we had three groups (non-frail, pre-frail and frail) and therefore we had to study two by two comparisons of each of the frailty stages (pre-frail vs non-frail, frail vs non-frail and frail vs pre-frail). Significant differences were found between the pre-frail and the non-frail, with the pre-frail being older, with a greater number of comorbidities and lower incomes. Living alone was non-significant, though the non-frail older patients cohabited more frequently in comparison with the other groups. These same factors were seen when comparing frail and non-frail participants, with the addition of a greater proportion of women in the frail group. Finally, analysis of the differences between the frail and the pre-frail individuals determined that the frail individuals were older, more likely to be women and to have a greater number of comorbidities. After summarizing all the results, we can say that income was the only factor that showed no variation, with an association in the pre-frail and frail groups only when compared with the non-frail older group. The rest of the factors changed as the frailty stage increased.

## DISCUSSION

### Summary

Comparison between pre-frail and frail older individuals shows that the probability of frailty increases as total comorbidities and age increase. Similarly, there was a higher proportion of women in the most advanced frailty stage. While no differences in income were seen between the pre-frail and frail older individuals, the non-frail group had higher incomes than the other groups, and income is thus a protective factor.

### Limitations

Concerning the potential limitations of our study, there may have been selection bias as sampling was consecutive not random, and the sample was taken from the population attending the healthcare center. The results may therefore vary in the general population. Information bias was minimized by using validated questionnaires, analyzing mainly objective variables and ensuring very rigorous data collection, obtaining all data in a

primary manner and not through retrospective clinical records. To minimize confounding bias, a multivariate analysis was performed using a model that considers the three frailty stages, which provides better information than assessing frailty alone. Moreover, we analyzed a selective primary care sample, instead of using a random selection, so the participants may not be representative of the population. Nevertheless, we should bear in mind that most patients of this age have chronic disorders and should therefore attend their healthcare center (place where they are treated in Spain) for their physician to control their disorders and prescribe or adapt the medication. Additionally, the number of factors for which associations with frailty was assessed could be limited. Consequently, it would be interesting to address other factors in future studies. We used the stage of frailty as a variable with three categories rather than just binary, thereby providing more information. This is important as, when applying mathematical models, no information that is available should be omitted, with quantitative variables to study their functional form and with qualitative variables not to create groups to dichotomize (*Moons et al., 2014*). Moreover, not having the frailty phenotype data available with performance measures could be a limitation of the current study, as we used the FRAIL scale. Finally, it would be interesting to obtain demographic characteristics in order to provide a greater chance of finding predictive factors that helped better predict frailty status. For example, variables related to lifetime and recent physical activity, as well as nutritional, alcohol and smoking behaviors would have been useful to include. Additionally, life transitions such as retirement, diagnosis of a major medical condition or moving from one type of residence to another (even from a house to an apartment) might have been useful to assess.

## Comparison with the existing literature

Concerning the four works that analyzed frailty as an ordinal variable, this measure in three cases was the Phenotype model, with the study in China using, like us, the FRAIL scale (*Curcio, Henao & Gomez, 2014*; *Hanlon et al., 2018*; *Ye, Gao & Fu, 2018*; *Siriwardhana et al., 2019*). Although using the same scale is better when making comparisons, we shall nevertheless attempt to undertake this comparison assuming the limitations.

Regarding the prevalence of the three possible frailty stages, 45.9% of the participants in our study were non-frail. This proportion was higher (59%) in the study conducted in the UK. However, the UK study sample included individuals aged ≥37 years, which explains these differences (*Hanlon et al., 2018*). Conversely, the pre-frailty percentage found in this study was similar to our findings and to those of the study from China (*Ye, Gao & Fu, 2018*; *Siriwardhana et al., 2019*). In the case of Sri Lanka and Colombia, though, pre-frailty increased because the studies were conducted in rural areas where there is usually greater pre-frailty and frailty. Finally, the prevalence of frailty in our sample was 20.3%, very similar to the values of the studies from Sri Lanka, Colombia and China (*Curcio, Henao & Gomez, 2014*; *Siriwardhana et al., 2019*; *Ye, Gao & Fu, 2018*), but considerably higher than the UK study (3%), probably due to the age cut-off point as a selection criterion (*Hanlon et al., 2018*).

Analysis of the factors associated with the highest frailty stage showed that, unlike in our study, the Chinese study did not compare by groups (pre-frail vs non-frail, frail vs non-frail) and assumed that the factors associated with frailty follow a natural order (ordinal model) (*Ye, Gao & Fu, 2018*), whereas the studies from the UK, Colombia and Sri Lanka (*Curcio, Henao & Gomez, 2014*; *Hanlon et al., 2018*; *Siriwardhana et al., 2019*), together with our study, use multinomial logistic regression. In addition, the Chinese study evaluated mainly psychosocial factors (*Ye, Gao & Fu, 2018*), which we have not addressed, making it difficult to compare our results with those of this study.

Among the factors studied that were statistically significant in several of the three comparisons (pre-frail vs non-frail, frail vs non-frail, pre-frail vs frail), we found differences in age, sex and income level, as well as in the number of comorbidities. Concerning age and sex, our results are entirely consistent with those of the Chinese study, finding a higher frailty stage with increasing age and in women (*Ye, Gao & Fu, 2018*). By contrast, the Sri Lankan study found that only the frail group was older than the non-frail group, with a very strong association from 75 years of age and with very relevant odds ratios (*Siriwardhana et al., 2019*). However, there were no significant differences in the increase in the number of women in the pre-frail or frail groups compared to the non-frail group (*Siriwardhana et al., 2019*). In the UK study, the results for age in the pre-frail vs non-frail group are similar and significant, while between the frail and non-frail groups there seems to be a direct association with age, that is, frailty increases with age. Regarding sex, there were more women in both the pre-frail and frail groups than in the non-frail group. Nevertheless, we should bear in mind that this study did not contemplate only older individuals (age ≥ 37 years) (*Hanlon et al., 2018*). Finally, the results of the Colombia study are fully in agreement with those of our work, as we too found a greater proportion of women and a higher age as the stage of frailty increased (*Curcio, Henao & Gomez, 2014*). When we examine the four studies together, there appears to be a clear association with age in the frail group, but the other associations change according to population, which could be due to sociocultural differences.

Another factor studied was income. In the Chinese study, income was not directly assessed, although it seems that the higher the level of education, the lower the frailty stage, and this can therefore be indirectly linked to income (*Ye, Gao & Fu, 2018*). In the UK, income was clearly associated with a lower socio-economic level and a higher prevalence of pre-frailty and frailty vs non-frailty (*Hanlon et al., 2018*). Finally, in the Sri Lankan study, the results concerning income did not reach significance, although an increasing trend was seen between higher frailty stage and lower income (*Siriwardhana et al., 2019*). In summary, an inverse association between income level and a higher frailty stage seems to be indicated. The final factor assessed was comorbidities, which were only evaluated in the UK and Colombia studies. The former showed that pre-frailty and, even more so, frailty are associated with multimorbidity (*Hanlon et al., 2018*), as in our study. However, the Columbia study only found differences in this aspect between the frail and the non-frail groups (*Curcio, Henao & Gomez, 2014*). This discrepancy may be because this study grouped the number of comorbidities in a binary form, establishing a cut point of ≥3, which could result in lack of significance.

Those studies that just assessed trends (*Abizanda et al., 2011*; *Hoogendijk et al., 2016*; *Langholz et al., 2018*; *Jacobsen et al., 2019*) mainly used the Fried phenotype. Comparison of their results with the factors also used by us shows total concordance for age, female sex and the number of comorbidities. These studies also assessed various other factors, such as educational level, which may be related to income, obtaining a similar result to ours. However, other factors studied included clinical and social conditions (attending a healthcare center or hospital, feeling lonely, self-perceived health…), which are not comparable to the factors evaluated in our study (*Abizanda et al., 2011*; *Hoogendijk et al., 2016*; *Langholz et al., 2018*; *Jacobsen et al., 2019*).

### Implications

This cross-sectional study found differences between non-frail, pre-frail and frail older adults in sex, age, number of disorders and income. This design opens up other research lines based on our results. The first practical implication concerns screening for frailty. As persons aged 60 years or over who have one or more factor found in this study have a higher risk of being frail (woman, older, a higher number of comorbidities and lower income), these persons should be assessed for frailty in primary care. This could enable it to be detected early or even reverse its stage (mainly from frail to pre-frail). Secondly, interventions could be considered based on preventive activities in older persons, focusing particularly on those persons who have some of the factors found here. The aim of these interventions would be to reduce the likelihood of frailty-associated complications (*Rockwood et al., 1994*; *Whitson, Purser & Cohen, 2007*; *Dent, Kowal & Hoogendijk, 2016*). Together, these two aspects would result in a better quality of life for older persons attending primary care centers and possibly reduce health care costs.

## CONCLUSIONS

Among people aged 60 years and over who seek primary care, differences were found in sex, age, income and the number of comorbidities between pre-frail and frail individuals. There was more likelihood of being in the frail group if the individual was a woman, older, had a higher number of comorbidities and lower income. Consequently, it is of great importance to promote the detection of these conditions so as to prevent frailty early.

## ACKNOWLEDGEMENTS

The authors thank Maria Repice and Ian Johnstone for their help with the English version of the text.

### Funding

The authors received no funding for this work.

### Competing Interests

Antonio Palazón-Bru is an Academic Editor for PeerJ.

## Author Contributions

- Vanessa Aznar-Tortonda conceived and designed the experiments, performed the experiments, prepared figures and/or tables, authored or reviewed drafts of the paper, and approved the final draft.
- Antonio Palazón-Bru conceived and designed the experiments, analyzed the data, prepared figures and/or tables, authored or reviewed drafts of the paper, and approved the final draft.
- Vicente Francisco Gil-Guillén conceived and designed the experiments, authored or reviewed drafts of the paper, and approved the final draft.

## Human Ethics

The following information was supplied relating to ethical approvals (i.e., approving body and any reference numbers):

The Clinical Research Ethics Committee of Sagunto Hospital and Elda University General Hospital approved the study on March 6, 2017.

## Data Availability

The data is available as a Supplemental File.

## Supplemental Information

Supplemental information for this article can be found online at http://dx.doi.org/10.7717/peerj.10380#supplemental-information.

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
