# Peer review of "Using the FRAIL scale to compare pre-existing demographic lifestyle and medical risk factors between non-frail, pre-frail and frail older adults accessing primary health care: a cross-sectional study"

_PeerJ, doi:10.7717/peerj.10380_

## Round 0.1 · original submission · Major Revisions

While two reviewers and I both see some merit in this study, there are also a number of major and minor concerns that you will need to address if this paper is to be more seriously considered for publication in PeerJ.

·

Basic reporting

1) I would encourage the authors to redo their literature search. Although the comparison of the three groups (non-frail, pre-frail, and frailty) is not often the primary focus of studies, there are dozens of studies that do report and statistically compare the characteristics of the three groups. Just like the current study does. For example, Hoogendijk et al. 2016 Maturitas, 83, 45-50; Abizanda et al 2011 JAGS, 59, 1356-1359. To mention just a few randomly. I am sure if the authors do a better search they will find 100s of papers.
2) It would also be good to provide a bit more context of frailty in the introduction. For example, use the recent frailty series in The Lancet (https://www.thelancet.com/series/frailty) or the Clegg 2013 Lancet seminar.
3) Please avoid the suggestion of longitudinal data. For example: “age, percentage of women and comborbidities increase as frailty progresses” (abstract) is too much suggestive of longitudinal data. Cross-sectional studies are not able to show anything about increase or progress. Please change this across the manuscript. Explanation: the higher percentage women and comorbidity in frail groups may be the result of the current sample selection (primary care visitors), and may say nothing about progress in frailty states.
4) The title of the manuscript is non-informative. It is not a continuum that is being assessed in this paper, but three categories. Better would be to make a title that covers the content of the study, for example:
5) Please use “older adults” or “older people” instead of “elderly”. The term “elderly” is nowadays regularly regarded as ageism.
6) What is “economic income”. Do the authors just mean “income”?

Experimental design

1) In the introduction, the authors state that the frailty phenotype (Fried 2001) is measured by the FRAIL scale. This is a wrong assumption. The FRAIL Scale is a questionnaire, with different items than the frailty phenotype. So, I assume the authors were confused? The frailty phenotype is mainly based on clinical performance measurements, and is not the same as the FRAIL scale. This mistake has great implications of the current study, as this requires a major revision throughout the manuscript. Not having the frailty phenotype data available with performance measures is a big limitation of the current study.
2) Sample selection: From the description of the abstract and the Methods section of the manuscript, it does not really become clear what kind of sample this is. It should be clearly stated that this is a selective primary care sample. There is no random sample selection, and the participants are not representative of a certain population. That limits the generalizability of the findings of this study. For example, it would be different if all primary care patients in a certain area would have been invited for this study. Now it only includes people that already go to the primary care practice with health problems. Therefore, the distribution may be different than in the general older population. This is perhaps why the frailty prevalence rather high for a 60+ population. Please also provide information on non-response, and reasons for non-response (refusal, missing data etc).
3) What are the implications of having a smaller sample size than needed according to the sample size calculation? Please also provide details on what type of sample size calculation was done?
4) The number of factors for which associations with frailty was assessed it rather limited. Why were only age, sex, partner status, income and comorbidity assessed? Would it not be a better contribution to the literature if a broader range of factors was assessed?
5) Primary care is not very strong in Spain. People with acute health problems often go directly to the emergency department in hospitals. This makes it difficult to judge the relevance of the current design. Is primary care in Spain an appropriate place to assess frailty? Please elaborate.
6) For variables that were not normally distributed, median (IQR) instead of mean (SD) should be reported.
7) Avoid terms like “nearly statistical significant” (Line 133). Non-significant is non-significant. Please interpret effect size, as the effect size could still be of value sometimes

Validity of the findings

1) The description of implications (line 211 etc) goes beyond the findings of the current study. This is a simple cross-sectional study on associations between some factors and frailty. So, please describe the implications that are more close to the findings of the current study. What do these associations tell us? Don`t talk about social relations, when these were not part of the current analysis (social isolation is much more than partner status).
2) The authors state that the multivariate model has good predictive power (line 222-227). This is wrong! The authors did not develop a prediction model, and did not assess predictive power. That is a completely different statistical approach. Also, variables were pre-selected by the authors, so we do not know anything about whether the chosen variables (age, sex, multimorbidity etc.) provide an optimal model. Please remove this complete paragraph. Also, the sentences on nutritional support come out of nowhere. Please stay more close to the content of the current study.
3) The interpretation of findings is not always accurate. Without clearly stating the comparison group, the findings have no meaning. For example “income is the only constant factor, with an association only with the robust group” (abstract) has no meaning if we don`t know what the comparison group is.
4) The reason that pre-frailty is not often studied, is that it is a poorly defined group. Some of these people will never become frail. And the group contains people that suffer from other conditions, like depression (because of the exhaustion question in the frailty measure). So how useful is it to focus on this group? This could be reflected on in the Discussion.
5) Please add the reference group to all findings, and do this in a consequent way. Now it is unclear what the results exactly mean (Results section), as the reference group is often not mentioned.
6) A bit more realistic overview of limitations should be provided. Now all limitations are directly presented as strengths. The authors should be a bit more cautious. This is a study with a selective, cross-sectional, sample and a limited data collection. So, calling this a “very rigorous data collection” is maybe a bit overstated (line 162). Another example, the mention of confounding bias (line 163) does not make any sense. Why would an analysis with three categories have less issues of confounding than an analysis of a binary variable? That has nothing to do with the number of categories, but with the appropriateness of the analytical model.
7) Mentioning the sample size as strong point is not always advisable (line 156). What does a sample size mean if you don`t have the right data? Quality above quantity. Moreover, a sufficient sample size is a prerequisite of performing a study, not a strength!
8) The comparison with previous studies is limited to very few studies. See my point above: the authors should do a better literature search, and compare with more findings from the European area (even the SHARE study has these data available).
9) The authors should acknowledge that they used a different frailty measure (FRAIL scale) than the studies they compare their findings with. The best would be to compare only with studies that really use the FRAIL scale.
10) The main conclusion is non-informative. What does “These factors behaved differently in the three groups, and are of great importance in promoting the detection of factors that characterize the non-frail, pre-frail and frail groups” mean? An unclear sentence. Please change this sentence. Just say something like: Being able to characterize the three groups may help to identify older adults at different frailty stages.

Additional comments

Thank you for the opportunity to review this interesting paper. While the paper is generally well-written, the implications of this cross-sectional descriptive study are somewhat limited. The paper is rather short. Please see my detailed comments above.

Reviewer 2 ·

Basic reporting

Overall a well conducted study with some flaws

1) There is widespread confusing use of both "robust" and "non-frail" to describe one group of study participants. For clarity, the authors should choose one term consistently throughout.

Similarly, in the abstract, it is stated that the study is "measuring robustness". This is erroneous. There is no measurement or quantification of robustness in the already robust population.

I propose the authors abandon the terms "robust" and "robustness" altogether.

2) Similarly, the following sentences are confusingly written in the abstract: "Concerning robustness, the pre-frail individuals were older, with more comorbidities and a lower economic income. Frail and robust participants showed the same differences, except that there was a higher proportion of women in the frail group. Among the frail and pre-frail individuals, the frail individuals were determined to be older, female, and with a greater number of comorbidities."

I suggest, for much more clarity:

"Compared to non-frail individuals, pre-frail individuals were older, with more comorbidities and a lower economic income. Compared to non-frail individuals, frail individuals were more likely to be female, older, with more comorbidities and a lower economic income. Compared to pre-frail individuals, frail individuals were more likely to be female, older and with more comorbidities."

3) The aims of the study are weakly described in lines 55-58. Instead of "factors", I would suggest "pre-existing demographic, lifestyle and medical risk factors". Furthermore, the specific subgroup comparison between pre-frail and frail individuals should be more explicitly highlighted in the aims (especially given the emphasis in the introduction, and in the background section of the abstract, that this comparison has been rarely performed).

4) In line 90-91, the phrases "and the findings of these studies are confirmed with our data. These variables were obtained by personal interview" are redundant and should be deleted.

5) in line 135, "exception" should be "addition".

6) in line 137, "women" should be "more likely to be women"

Experimental design

1) Somewhere in the statistics section of the methods, it should be explained that in the multivariate logistic regression, income was analysed as a qualitative variable with the lowest income bracket being set as the reference point.

2) Furthermore, the statistical methods used in the "bivariate analysis" in Table 1 are not described in the statistics section of the methods. Is this using Chi-squared and Students T test analysis?

Validity of the findings

1) Table 1 should also include p values for the between-paired group comparisons, just like Table 2, rather than single omnibus p-values between all three groups. This would be more in keeping with the implicit study aim to determine differences between the pre-frail and frail groups.

2) The summary paragraph in lines 144-149 should also be reworded to focus more strongly on the specific subgroup comparison between pre-frail and frail individuals (especially given the emphasis in the introduction, and in the background section of the abstract, that this comparison has been rarely performed).

3) The conclusions in lines 135-139 section is very weak. Again, it should be reworded to focus more strongly on the specific subgroup comparison between pre-frail and frail individuals. The phrase "these factors behaved differently in the three groups" is inadequate: the conclusion should reiterate the directions and magnitude/significance of these differences.

---

## Round 0.2 · Minor Revisions

General comments
Thanks for attending to most of the comments of the two reviewers. Further ways that the manuscript needs to be improved prior to be accepted for publication are described below. When I refer to a particular line, I’m referring to the track changes Word document.

Specific comments:

Title: the title is better now but it might read even more accurately as “Using the FRAIL scale to compare pre-existing demographic lifestyle and medical risk factors between non-frail, pre-frail and frail older adults accessing primary health care: a cross-sectional study”.

Keyword: remove frail elderly based on the connotation of this term as ageism.

Line 128 – 142: I wondered why other demographic characteristics were not obtained in order to provide a greater chance of finding predictive factors that helped better predict frailty status. For example, I would have thought some questions related to lifetime and recent physical activity, nutritional, alcohol and smoking behaviours would have been useful to include. Further, could life transitions such as retirement, diagnosis of major medical condition or moving from one type of residence to another (even from a house to an apartment) have been useful to assess?

Line 308 – 315: I’m not sure how sex, age, number of disorders and income open up other lines of research or how it will allow you to establish a prediction model in relation to transition from one level of frailty to another. Please reconsider the strength of the arguments in this paragraph what are its potential applications to maximising the public health benefit at minimal cost with respect to screening (assessment) and interventions for frailty for older adults accessing primary health care.

Reviewer 2 ·

Basic reporting

All my previous criticisms have been answered, thank you.

Experimental design

All my previous criticisms have been answered, thank you.

Validity of the findings

All my previous criticisms have been answered, thank you.

---

## Round 0.3 · accepted · Accept

I think the authors for their hard work in attending to the comments of the reviewers and I. I would therefore like to recommend this paper be accepted for publication in PeerJ.